# Exploring the Efficacy of Alpha-Lipoic Acid in Comorbid Osteoarthritis and Type 2 Diabetes Mellitus

**DOI:** 10.3390/nu16193349

**Published:** 2024-10-02

**Authors:** Iryna Halabitska, Valentyn Oksenych, Oleksandr Kamyshnyi

**Affiliations:** 1Department of Therapy and Family Medicine, I. Horbachevsky Ternopil National Medical University, 46001 Ternopil, Ukraine; 2Broegelmann Research Laboratory, Department of Clinical Science, University of Bergen, 5020 Bergen, Norway; 3Department of Microbiology, Virology and Immunology, I. Horbachevsky Ternopil National Medical University, 46001 Ternopil, Ukraine; kamyshnyi_om@tdmu.edu.ua

**Keywords:** osteoarthritis, type 2 diabetes, diabetes mellitus, alpha-lipoic acid, inflammation, oxidative stress, glycemic control, comorbidity

## Abstract

**Background/Objectives.** The comorbidity of osteoarthritis and type 2 diabetes mellitus poses a complex clinical challenge, complicating patient management due to overlapping pathophysiological mechanisms. This research aims to analyze the exacerbation of clinical symptoms and biochemical markers in patients with OA and T2DM compared to those with OA alone. **Methods**. We employed various assessment methods to evaluate inflammation, oxidative stress, and glycemic control in both cohorts. This study includes the administration of alpha-lipoic acid (ALA) to patients with comorbid OA and T2DM, monitoring its effects on joint function, inflammatory markers, oxidative stress levels, and glycemic control. **Results.** The findings indicate that T2DM significantly worsens clinical symptoms and biochemical markers in OA patients. Those with both conditions exhibited elevated indicators of inflammation and oxidative stress compared to OA-only patients. Additionally, correlations among metabolic, psychological, and inflammatory factors were identified. Body mass index emerged as a potential predictor for the deterioration of evaluated parameters. The analysis revealed that ALA administration led to statistically significant improvements in WOMAC pain scores, the Lequesne Algofunctional Index, and the AIMS-P compared to the control group. **Conclusions.** Further research into ALA’s effects on OA progression in patients with comorbidities is essential for developing personalized treatment approaches.

## 1. Introduction

With a global prevalence of nearly 23%, osteoarthritis (OA) imposes a significant disease burden, exacerbated by the lack of effective disease-modifying treatments [1,2]. Affecting over 240 million people globally, OA is a major contributor to activity limitation among adults. The pathogenesis of OA remains poorly understood, leading to a lack of effective clinical treatments [3]. Disruptions in glucose metabolism have been shown to cause chondrocyte hypertrophy and extracellular matrix degradation, playing a significant role in the onset and progression of the disease [4]. OA is a degenerative joint disorder that primarily targets the articular cartilage and is influenced by factors such as trauma, metabolic processes, and comorbidities [5]. The ensuing chronic inflammation affects not only the cartilage, but also adjacent tissues, further impairing function and exacerbating OA symptoms [5]. The prevalence of osteoarthritis rises with age, and its histological and pathophysiological characteristics indicate an underlying inflammatory process [6]. OA and diabetes mellitus (DM) are both prevalent chronic disorders with long-term detrimental effects. The pathophysiology of OA is significantly influenced by DM and its associated factors, including hyperglycemia [7].

DM is a significant chronic condition characterized by inadequate insulin production or the body’s inability to effectively use insulin, leading to a long-term metabolic disorder [8]. Currently, it affects approximately 537 million adults worldwide, or 10.5% of those aged from 20 to 79. Projections indicate that this number will rise to 643 million by 2030 and reach 783 million by 2045 [9]. Several risk factors, such as severe obesity, a family history of diabetes, certain ethnic backgrounds, maternal diabetes or gestational diabetes, and being female, contribute to the earlier onset of type 2 diabetes mellitus (T2DM) [8,10]. Pathogenetically, the primary mechanism involves impaired insulin secretion, coupled with insulin resistance related to ectopic fat accumulation [11,12]. The comorbidity of T2DM and OA is quite common, given that both conditions are prevalent metabolic diseases sharing risk factors such as obesity and aging. Despite this frequent coexistence, there is no clear consensus regarding the direct impact of T2DM on the development and progression of OA. Research findings have been inconsistent, with some studies demonstrating a significant influence of DM on OA, while others have found no substantial link between the two conditions [13,14]. Alpha-lipoic acid (ALA) is used in the comprehensive management of diabetes mellitus due to its antioxidant and pro-oxidant properties that affect insulin sensitivity and secretion [15]. It is widely prescribed for diabetic polyneuropathy, where it enhances nerve conduction and alleviates symptoms. Furthermore, ALA is also utilized in the treatment of other insulin resistance conditions, such as metabolic syndrome, polycystic ovary syndrome, and obesity [16,17,18]. Studies show that ALA significantly benefits the management of osteoarthritis in patients with DM [19,20,21]. The use of ALA for managing patients with comorbid osteoarthritis and type 2 diabetes mellitus requires further exploration. Continued research is essential to elucidate its effectiveness and potential advantages in this particular cohort.

## 2. Materials and Methods

### 2.1. Sample Collection

Participants for this study were recruited from the Ternopil City Communal Institution “Center for Primary Medical and Sanitary Care” between 2020 and 2023. The research adhered to the core principles set forth in the Council of Europe’s Convention on Human Rights and Biomedicine and was conducted in accordance with the ethical guidelines specified in the Declaration of Helsinki by the World Medical Association, including any subsequent revisions. Additionally, the study complied with Ministry of Health of Ukraine Order No. 690 dated 23 September 2009. Informed consent was obtained from all participants prior to their involvement. The study received ethical approval from the Bioethics Committee of I. Horbachevsky Ternopil National Medical University Ministry of Health of Ukraine (Protocol No. 75, 1 November 2023).

The study cohort comprised individuals of Ukrainian ethnicity with European ancestry aged between 29 and 78 years.

The study involved 123 patients who were categorized according to the specific pathology under investigation and the presence of comorbidities. The first group consisted of 52 patients diagnosed with OA, while the second group comprised 71 patients with concurrent OA and T2DM. No notable variations in gender, age, or the duration of OA were detected in the study groups (Table 1).

The cohort of patients with OA and T2DM was subsequently divided according to the treatment regimen: one group without ALA (*n* = 37) and another group with ALA (*n* = 34). No significant differences in gender, age, duration of OA, or duration of T2DM were observed among the study groups (Table 2).

The group comprising patients with comorbid osteoarthritis and type 2 diabetes was segmented into two subgroups to facilitate a more in-depth examination of the potential effects of alpha-lipoic acid on the investigated parameters.

The group not receiving ALA was treated according to standard protocols, which included a regimen of basic therapy medications, such as nonsteroidal anti-inflammatory drugs, chondroprotectors, and oral hypoglycemic agents (metformin). The group receiving ALA, in addition to the basic therapy regimen, was administered alpha-lipoic acid by oral administration at a dose of 600 mg once daily for 6 weeks. The levels of the assessed parameters were measured before treatment and after 6 weeks.

Inclusion criteria for the study encompassed individuals of both genders who had a confirmed diagnosis of hip and knee osteoarthritis (ICD-10 codes M16 and M17) and type 2 diabetes mellitus (ICD-10 code E11). Exclusion criteria included: type I diabetes mellitus; thyroid gland disorders, decompensated heart-lung diseases, acute myocardial infarction; stage II-III hypertension, arrhythmias, unstable ischemic heart disease, recent major surgery within the past month; stage III-V chronic kidney disease; use of systemic glucocorticosteroids; severe exhaustion, pregnancy, suspected malignant tumors, infectious and parasitic diseases, congenital anomalies and chromosomal disorders, bleeding tendencies, psychiatric and behavioral disorders, and refusal to participate in the study.

The diagnosis of OA was made based on international guidelines [22]. The assessment of the radiological stages of OA were classified according to the system developed by J.H. Kellgren and J.S. Lawrence, with magnetic resonance imaging (MRI) findings also incorporated into the analysis, and standardized clinical criteria. The diagnosis of T2DM was confirmed in accordance with international standards, based on elevated serum glucose levels and HbA1c values, in accordance with the guidelines set forth by the American Diabetes Association [23].

### 2.2. Laboratory and Clinical Data

In assessing joint status in individuals with OA, the Western Ontario and McMaster Universities Osteoarthritis Index (WOMAC) was utilized as a key evaluation tool [24]. To gauge the severity of OA within the study cohort, the Lequesne Algofunctional Index was applied [24]. Additionally, the Visual Analog Scale (VAS) was used to measure rest pain, movement pain, inflammation, and joint dysfunction [25]. The Arthritis Impact Measurement Scales (AIMS) were employed to assess the effect of arthritis on patients’ physical and emotional health, measuring multiple dimensions such as Physical Function (FF), Pain (P), Social Function (SF), Emotional Health (EH), and General Health Perception (GHP) [26].

Fasting blood glucose levels were measured using automated glucose oxidase methods, while C-peptide levels were analyzed via enzyme-linked immunosorbent assay (ELISA) (BIOSERV Diagnostics Gmbh (Rostock, Germany)). HOMA-IR was calculated from fasting insulin and glucose levels determined using a chemiluminescent immunoassay, and Hemoglobin A1c (HbA1c) levels were assessed using high-performance liquid chromatography (HPLC).

The Diabetes Distress Scale (DDS-17) was used to evaluate various dimensions of distress related to diabetes management, including emotional burden (DDS-17-EB), physician-related distress (DDS-17-PRD), regimen-related distress (DDS-17-RRD), interpersonal distress (DDS-17-ID), and Total DDS-17 score [27]. Additionally, the Problem Areas in Diabetes (PAID) scale assessed overall emotional distress associated with diabetes through a self-administered questionnaire [28].

Leukocytes, neutrophils, and lymphocytes were quantified using automated complete blood count (CBC) analyzers. The neutrophil-to-lymphocyte ratio (NLR) was calculated from these CBC results. C-Reactive Protein (CRP) levels were measured with ELISA (BIOSERV Diagnostics Gmbh (Germany)). Hydroxyproline and malonaldehyde (MA) were quantified using colorimetric assays and thiobarbituric acid reactive substances (TBARS) assays. Ceruloplasmin, α_1_-antitrypsin, and α_2_-Macroglobulin levels were determined using nephelometry or ELISA (BIOSERV Diagnostics Gmbh (Germany)), while kallikrein levels were assessed through enzyme immunoassays. Superoxide dismutase (SOD) and catalase activities were measured using colorimetric and spectrophotometric assays.

Serum IgA, IgM, IgG, and IgE levels were measured using standard immunoassays, including nephelometry for IgA, IgM, and IgG, and ELISA for IgE (BIOSERV Diagnostics Gmbh (Germany)). T-lymphocytes (CD3+, CD19−), T-helpers (CD4+, CD8−), T-cytotoxic cells (CD4−, CD8+), cytotoxic cells (CD3+, CD56+), NK cells (CD3−, CD56+), and B-lymphocytes (CD3−, CD19+) were quantified using flow cytometry. The immunoregulatory index was calculated based on the proportions of T-helpers, T-cytotoxic cells. Monocytes/macrophages (CD14) were also assessed via flow cytometry using CD14-specific antibodies.

The methodology for assessing the body mass index (BMI) involved calculating the ratio of weight in kilograms to the square of height in meters (kg/m^2^) for all participants, ensuring accurate measurements through standardized protocols for weight and height assessment.

### 2.3. Statistical Analysis

Patient demographics and clinical data were comprehensively analyzed and presented using descriptive statistics. The Shapiro–Wilk test was employed to evaluate the normality of data distribution. Given the non-normal distribution of the data, medians and interquartile ranges were calculated for all variables, and a significance level of *p* < 0.05 was used for hypothesis testing.

Differences between two independent groups were assessed using the Mann–Whitney U test, while the Kruskal–Wallis test was used for comparisons involving three or more groups, with Dunn’s multiple comparison test applied for pairwise differences in a post hoc analysis.

Spearman’s rank correlation coefficient was utilized to explore associations among continuous variables in the correlation matrix analysis.

Binary logistic regression was utilized to identify potential predictors linked to the comorbidity of OA and T2DM. The model quality analysis included the creation of an odds plot and the calculation of ROC metrics, such as the Area Under the ROC Curve (AUC) with a 95% confidence interval (CI), the Youden index (J) with its corresponding cutoff point, as well as sensitivity (Se) and specificity (Sp).

The Wilcoxon signed-rank test was utilized to assess the significance of differences within a group before and after treatment. To compare the differences between groups prior to treatment, the Mann–Whitney U test was employed. The Mann–Whitney U test was also applied to evaluate the differences between groups following treatment. The Difference-in-Differences (DiD) test was employed to assess the causal impact of the intervention by comparing outcome changes over time between the treatment and control groups.

Statistical analyses were performed using commercially available software packages, including IBM SPSS Statistics (version 25) and GraphPad Prism 10.1.1, Stata 16.

## 3. Results

### 3.1. Evaluation of Clinical, Biochemical, and Immunological Parameters in OA and OA + T2DM

Several of the examined indicators revealed statistically significant differences between the groups (Table 3). The OA + T2DM group presented with elevated levels in WOMAC stiffness, total WOMAC score, the Lequesne Algofunctional Index, VAS rest pain, VAS inflammation, VAS joint dysfunction, AIMS-FF, AIMS-SF, and AIMS-GHP (*p* < 0.05). These results indicate that the coexistence of T2DM with OA may intensify symptoms and contribute to more pronounced joint functional impairments.

A statistically significant elevation was observed in various indicators associated with the progression of type 2 diabetes among patients with comorbid OA and T2DM compared to those in the group with osteoarthritis. Specifically, the OA + T2DM group demonstrated higher levels in fasting blood glucose, C-peptide, HOMA-IR, HbA1c, DDS-17-EB, DDS-17-PRD, DDS-17-RRD, DDS-17-ID, total DDS-17 score, and PAID (*p* < 0.001) (Table 4).

Statistically significant differences were observed in several biomarkers between the OA and OA + T2DM groups. Specifically, the OA + T2DM group had significantly higher levels of neutrophils, NLR, MA, ceruloplasmin, and catalase (*p* < 0.05) (Table 5). These findings indicate an exacerbating effect of comorbid type 2 diabetes mellitus on the deterioration of these parameters in osteoarthritis. The increased levels of inflammatory and oxidative stress markers suggest that the presence of T2DM may contribute to a more severe pathological profile and worsening of the clinical status in patients with OA.

In the analysis of immunological parameters, significant differences were detected between the OA and OA + T2DM groups (Table 6). Serum IgA and IgE levels were notably higher in the OA + T2DM group (*p* < 0.05). Additionally, the Immunoregulatory Index was significantly elevated in the OA + T2DM group (*p* < 0.05). No significant differences were observed for IgM, IgG, T-Lymphocytes, T-Helpers, T-Cytotoxic Cells, Cytotoxic Cells, NK Cells, B-Lymphocytes, or Monocytes/Macrophages (*p* > 0.05).

An analysis of BMI was also conducted in the studied groups. In the cohort of patients with osteoarthritis, the BMI was determined to be 27.85 kg/m^2^ (26.76–29.14 kg/m^2^), while in the cohort of patients with comorbidities, the BMI was 29.22 kg/m^2^ (27.04–31.32 kg/m^2^). A statistically significant difference between these values was observed (*p* = 0.0064), suggesting a potential influence of increased body weight on the studied parameters, which necessitates further investigation.

### 3.2. Correlation Analysis of Data in Patients with OA and OA + T2DM

This section explores the correlations among various clinical parameters in OA patients without T2DM. Spearman’s rank correlation coefficient (r) was employed to assess the strength and direction of these relationships. The total WOMAC score exhibited positive and negative correlations with several parameters. These correlations included the Lequesne Algofunctional Index (r = 0.36, *p* = 0.009), AIMS-GHP (r = 0.64, *p* < 0.001), HOMA-IR (r = 0.37, *p* = 0.006), HbA1c (r = 0.31, *p* = 0.023), total DDS-17 score (r = 0.41, *p* = 0.002), SOD (r = −0.64, *p* < 0.001), MA (r = 0.35, *p* = 0.011), ceruloplasmin (r = 0.46, *p* = 0.001), IgE (r = 0.3, *p* = 0.032), and the Immunoregulatory Index (r = 0.4, *p* = 0.003). The Lequesne Algofunctional Index exhibited positive and negative correlations with PAID (r = 0.29, *p* = 0.037), SOD (r = −0.51, *p* < 0.001), catalase (r = −0.36, *p* = 0.009), and the Immunoregulatory Index (r = 0.39, *p* = 0.004). VAS inflammation showed positive and negative correlations with total DDS-17 score (r = 0.31, *p* = 0.027), catalase (r = −0.36, *p* = 0.009), and B-Lymphocytes (CD3−. CD19+) (r = 0.45, *p* < 0.001). AIMS-GHP displayed positive and negative correlations with HOMA-IR (r = 0.39, *p* = 0.004), HbA1c (r = 0.45, *p* < 0.001), total DDS-17 score (r = 0.34, *p* = 0.015), SOD (r = −0.58, *p* < 0.001), CRP (r = 0.33, *p* = 0.017), MA (r = 0.64, *p* < 0.001), ceruloplasmin (r = 0.52, *p* < 0.001), IgE (r = 0.58, *p* < 0.001), and the Immunoregulatory Index (r = 0.34, *p* = 0.015). HOMA-IR exhibited positive and negative correlations with total DDS-17 score (r = 0.47, *p* < 0.001), SOD (r = −0.39, *p* = 0.005), NLR (r = 0.34, *p* = 0.013), MA (r = 0.47, *p* < 0.001), ceruloplasmin (r = 0.5, *p* < 0.001), catalase (r = −0.3, *p* = 0.033), and B-Lymphocytes CD3−. CD19+ (r = 0.52, *p* < 0.001). HbA1c demonstrated positive correlations with SOD (r = −0.33, *p* = 0.019), CRP (r = 0.29, *p* = 0.04), MA (r = 0.33, *p* = 0.016), and IgE (r = 0.42, *p* = 0.002). Total DDS-17 score demonstrated positive and negative correlations with MA (r = 0.41, *p* = 0.002), ceruloplasmin (r = 0.53, *p* < 0.001), catalase (r = −0.38, *p* = 0.006), and B-Lymphocytes (CD3−. CD19+) (r = 0.35, *p* = 0.011). SOD showed positive and negative correlations with CRP (r = −0.27, *p* = 0.049), MA (r = −0.44, *p* < 0.001), ceruloplasmin (r = −0.45, *p* < 0.001), catalase (r = 0.38, *p* = 0.006), IgE (r = −0.32, *p* = 0.002), and the Immunoregulatory Index (r = −0.42, *p* < 0.001). NLR exhibited positive and negative correlations with CRP (r = 0.47, *p* < 0.001), MA (r = 0.4, *p* = 0.004), catalase (r = −0.38, *p* = 0.005), the Immunoregulatory Index (r = 0.54, *p* < 0.001), and B-Lymphocytes (CD3−. CD19+) (r = 0.35, *p* = 0.012). CRP showed positive and negative correlations with MA (r = 0.4, *p* = 0.034), catalase (r = −0.38, *p* < 0.001), the Immunoregulatory Index (r = 0.59, *p* < 0.001), and B-Lymphocytes (CD3−. CD19+) (r = 0.35, *p* = 0.011). MA demonstrated positive correlations with ceruloplasmin (r = 0.4, *p* = 0.004) and B-Lymphocytes (CD3−. CD19+) (r = 0.54, *p* < 0.001). Ceruloplasmin exhibited positive correlations with IgE (r = 0.29, *p* = 0.04) and B-Lymphocytes (CD3−. CD19+) (r = 0.34, *p* = 0.015). Catalase showed a positive correlation with the Immunoregulatory Index (r = −0.48, *p* < 0.001).

The subsequent section examines the correlations among various clinical parameters in OA patients with T2DM. Spearman’s rank correlation coefficient (r) was used to determine the magnitude and direction of these associations. Total WOMAC score demonstrated positive and negative correlations with the Lequesne Algofunctional Index (r = 0.66, *p* < 0.001), VAS movement pain (r = 0.25, *p* = 0.034), AIMS-FF (r = 0.49, *p* < 0.001), AIMS-GHP (r = 0.62, *p* < 0.001), HOMA-IR (r = 0.27, *p* = 0.024), HbA1c (r = 0.36, *p* = 0.002), total DDS-17 score (r = 0.44, *p* < 0.001), PAID (r = 0.51, *p* < 0.001), NLR (r = 0.51, *p* < 0.001), ceruloplasmin (r = 0.27, *p* = 0.024), catalase (r = −0.25, *p* = 0.033), IgE (r = 0.44, *p* < 0.001), and the Immunoregulatory Index (r = 0.51, *p* < 0.001). The Lequesne Algofunctional Index showed positive and negative correlations with VAS inflammation (r = 0.35, *p* = 0.003), AIMS-GHP (r = 0.6, *p* < 0.001), HbA1c (r = 0.41, *p* < 0.001.), total DDS-17 score (r = 0.37, *p* = 0.001), PAID (r = 0.37, *p* = 0.002), NLR (r = 0.6, *p* < 0.001), CRP (r = 0.24, *p* = 0.041), MA (r = 0.36, *p* = 0.003), ceruloplasmin (r = 0.24, *p* = 0.044), catalase (r = −0.24, *p* = 0.043), IgE (r = 0.46, *p* < 0.001), and the Immunoregulatory Index (r = 0.41, *p* < 0.001). VAS movement pain demonstrated positive and negative correlations with HbA1c (r = 0.43, *p* < 0.001), total DDS-17 score (r = 0.27, *p* = 0.022), NLR (r = 0.34, *p* = 0.004), and catalase (r = −0.39, *p* < 0.001). VAS inflammation showed positive and negative correlations with AIMS-GHP (r = 0.27, *p* = 0.022), HOMA-IR (r = 0.26, *p* = 0.029), HbA1c (r = 0.31, *p* = 0.008), total DDS-17 score (r = 0.43, *p* < 0.001), SOD (r = −0.41, *p* < 0.001), NLR (r = 0.24, *p* = 0.048), MA (r = 0.37, *p* = 0.002), ceruloplasmin (r = 0.42, *p* < 0.001), and IgE (r = 0.41, *p* < 0.001.). AIMS-FF demonstrated positive correlations with AIMS-GHP (r = 0.55, *p* < 0.001), HOMA-IR (r = 0.35, *p* = 0.003), PAID (r = 0.45, *p* < 0.001), CRP (r = 0.38, *p* = 0.001), and the Immunoregulatory Index (r = 0.54, *p* < 0.001). AIMS-GHP showed positive and negative correlations with HbA1c (r = 0.31, *p* = 0.009), total DDS-17 score (r = 0.35, *p* = 0.003), PAID (r = 0.56, *p* < 0.001), NLR (r = 0.43, *p* < 0.001), CRP (r = 0.33, *p* = 0.005), MA (r = 0.29, *p* = 0.013), ceruloplasmin (r = 0.31, *p* = 0.008), catalase (r = −0.24, *p* = 0.045), IgE (r = 0.31, *p* = 0.008), and the Immunoregulatory Index (r = 0.58, *p* < 0.001). HOMA-IR exhibited positive correlations with HbA1c (r = 0.29, *p* = 0.013), PAID (r = 0.24, *p* = 0.041), and MA (r = 0.28, *p* = 0.019). HbA1c demonstrated positive and negative correlations with total DDS-17 score (r = 0.37, *p* = 0.002), NLR (r = 0.38, *p* < 0.001), MA (r = 0.33, *p* = 0.005), catalase (r = −0.47, *p* < 0.001), and IgE (r = 0.52, *p* < 0.001). Total DDS-17 score showed positive and negative correlations with SOD (r = −0.27, *p* = 0.024), NLR (r = 0.41, *p* < 0.001), MA (r = 0.35, *p* = 0.003), ceruloplasmin (r = 0.5, *p* < 0.001), catalase (r = −0.37, *p* = 0.002), IgE (r = 0.36, *p* = 0.002), and the Immunoregulatory Index (r = 0.31, *p* = 0.01). PAID demonstrated positive correlations with NLR (r = 0.45, *p* < 0.001) and the Immunoregulatory Index (r = 0.27, *p* = 0.025). SOD exhibited positive correlations with MA (r = −0.26, *p* = 0.029), ceruloplasmin (r = −0.33, *p* = 0.005), and IgE (r = −0.24, *p* = 0.041). NLR demonstrated positive and negative correlations with MA (r = 0.28, *p* = 0.017), catalase (r = −0.41, *p* < 0.001), and IgE (r = 0.37, *p* = 0.001). CRP exhibited a positive correlation with the Immunoregulatory Index (r = 0.62, *p* < 0.001). MA showed positive correlations with ceruloplasmin (r = 0.31, *p* = 0.008), IgE (r = 0.27, *p* = 0.022). Ceruloplasmin demonstrated a negative correlation with catalase (r = −0.24, *p* = 0.04).

### 3.3. Comparative Analysis of Treatment Outcomes in Patients with OA and T2DM with and without ALA Supplementation

A statistical evaluation of the parameters before and after treatment revealed significant changes (Table 7). In the group without ALA, notable improvements were observed in WOMAC pain, WOMAC stiffness, WOMAC Physical Function, Total WOMAC score, the Lequesne Algofunctional Index, VAS rest pain, VAS movement pain, VAS inflammation, VAS joint dysfunction, AIMS-FF, and AIMS-SF (*p* < 0.05). Conversely, the group with ALA exhibited significant enhancements in WOMAC pain, WOMAC Physical Function, Total WOMAC score, the Lequesne Algofunctional Index, VAS rest pain, VAS movement pain, VAS inflammation, VAS joint dysfunction, AIMS-FF, AIMS-P, AIMS-SF, AIMS-EH, and AIMS-GHP (*p* < 0.05). A comparative analysis between the groups after treatment highlighted that the effects of treatment were more pronounced in the ALA group, particularly for WOMAC pain, the Lequesne Algofunctional Index, VAS joint dysfunction, AIMS-FF, and AIMS-EH (*p* < 0.05) (Figure 1).

In the analysis of T2DM progression indicators before and after treatment (Table 8), significant improvements were observed across several parameters. In the group not receiving ALA, notable reductions were seen in fasting blood glucose levels, DDS-17-EB, DDS-17-RRD, and PAID (*p* < 0.05). In contrast, the group treated with ALA demonstrated significant improvements in fasting blood glucose levels, DDS-17-EB, DDS-17-RRD, DDS-17-ID, and PAID (*p* < 0.05). Post-treatment comparisons between the groups revealed that the ALA group experienced more substantial benefits, particularly in DDS-17-EB and PAID (*p* < 0.05) (Figure 2).

The analysis of outcomes before and after treatment revealed significant differences in inflammatory markers, oxidative stress parameters, and leukocyte subpopulations between the groups (Table 9). In the group not receiving ALA, statistically significant changes were observed in NLR, CRP, and catalase levels (*p* < 0.05). Conversely, in the group treated with ALA, significant alterations were noted in leukocyte counts, neutrophil percentages, NLR, CRP, MA, ceruloplasmin, SOD, and catalase (*p* < 0.05). Additionally, biomarkers after treatment comparisons between the groups highlighted a statistically significant difference in CRP levels (*p* < 0.05) (Figure 3).

The analysis indicated differential responses in immunological markers to treatment across the study groups (Table 10). In the group that did not receive ALA, no statistically significant changes were observed in the immunological markers (*p* > 0.05). In contrast, the group treated with ALA showed significant post-treatment changes in serum IgA, IgE, and the Immunoregulatory Index (*p* < 0.05). Moreover, a significant difference in IgE levels was observed when comparing post-treatment levels between the groups across the studied markers (*p* < 0.05) (Figure 4).

## 4. Discussion

This study sheds light on the intricate relationship between T2DM and OA, demonstrating how the coexistence of these conditions significantly exacerbates clinical symptoms and biochemical markers in affected patients. These findings are consistent with previous studies that have reported similar associations between T2DM and the worsening of OA [29,30,31,32]. For instance, prior research has shown that patients with both T2DM and OA often experience more severe pain and greater functional limitations compared to those with OA alone [14,33,34,35,36]. This study aligns with these observations, further supporting the notion that T2DM acts as an aggravating factor in OA progression.

The increased levels of inflammation, oxidative stress, and immune disturbances observed in this study’s cohort mirror findings from other research [10,37,38]. Elevated CRP levels, a marker of systemic inflammation, have been consistently reported in patients with T2DM [39,40], and their further elevation in the context of OA suggests the synergistic effect of these conditions on inflammatory processes [30,41]. Previous studies have also highlighted the role of oxidative stress in comorbid T2DM and OA [38,42], with some research indicating that oxidative damage may contribute to cartilage degradation and joint dysfunction [43,44,45]. The current study’s findings reinforce these insights, showing that oxidative stress is markedly higher in patients with comorbid T2DM and OA, thereby linking metabolic disturbances with joint pathology.

Moreover, this study’s identification of significant correlations among metabolic, psychological, and inflammatory factors in patients with both T2DM and OA adds a new dimension to our understanding of this comorbidity. While earlier research has explored the individual impact of these factors [34,46,47,48,49], the present study underscores their interconnectedness, suggesting that managing one aspect of this triad may influence the others. For example, addressing metabolic dysregulation in T2DM could potentially reduce inflammation and improve psychological well-being, which in turn might slow the progression of OA [50,51,52]. This holistic view aligns with recent studies advocating for integrated treatment approaches that consider the multifactorial nature of comorbid conditions [53,54,55]. T2DM is managed by targeting metabolic alterations, with metformin being a key pharmacological agent [56,57,58]. It effectively improves insulin sensitivity, decreases hepatic glucose production, and enhances peripheral glucose uptake, thereby aiding in the normalization of metabolic parameters in individuals with T2DM [59,60,61,62,63].

This study’s exploration of ALA as a therapeutic adjunct also contributes to the growing body of evidence supporting its use in metabolic disorders. Previous studies have demonstrated ALA’s effectiveness in improving insulin sensitivity and reducing oxidative stress (Figure 5) [64,65,66]. Recent research indicates that the administration of alpha-lipoic acid is associated with a significant reduction in systemic inflammation. This effect highlights its potential as a therapeutic agent in the management of inflammatory processes in various conditions (Figure 6) [67,68,69,70].

**Figure 5 nutrients-16-03349-f005:**
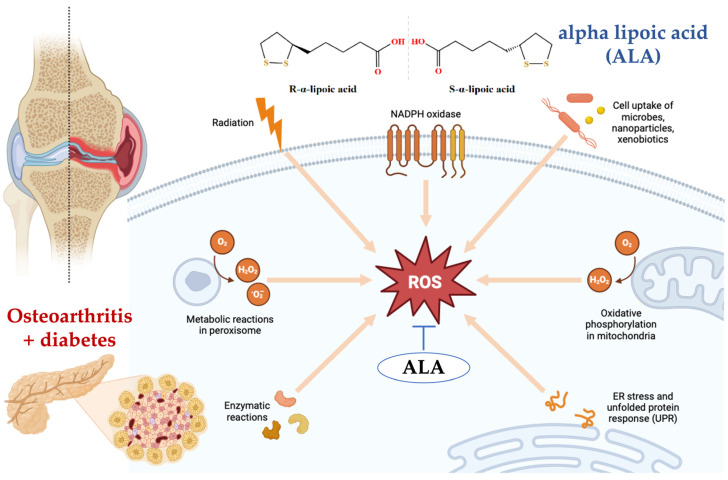
Alpha-lipoic acid (ALA) reduces the formation of reactive oxygen species (ROS) by directly scavenging free radicals, including hydroxyl radicals and hydrogen peroxide. Additionally, ALA chelates transition metal ions, preventing the conversion of weak oxidants into more harmful reactive species. ALA also enhances the levels of intracellular antioxidants such as glutathione, which further helps in neutralizing ROS. Moreover, ALA can regenerate other antioxidants like vitamin C and vitamin E, thereby maintaining a robust antioxidant defense system within the cell [71,72,73]. Figure 5 was generated using BioRender (https://www.biorender.com/, accessed on 12 August 2024).

The current research extends these findings by demonstrating that ALA supplementation may have a potential impact on the course of pain syndrome in osteoarthritis, as well as a possible improvement in glycemic control, potentially enhancing the effects of hypoglycemic therapy and diet in patients with comorbid OA T2DM. These results are consistent with other studies that have highlighted the potential of antioxidants in managing both metabolic and degenerative joint diseases [74,75,76].

However, the findings regarding ALA must be interpreted with caution. While the improvements observed in this study are promising, they are consistent with earlier reports that emphasize the need for further investigation into the long-term effects and optimal dosing of ALA in patients with multiple comorbidities [77,78,79]. Other studies have pointed out the variability in patient responses to ALA, which may depend on factors such as disease severity, duration of diabetes, and individual differences in metabolism [80,81,82,83]. The current study’s relatively short duration and limited sample size suggest that more extensive research is needed to confirm these preliminary results and to explore the potential mechanisms underlying ALA’s effects in this specific patient population.

**Figure 6 nutrients-16-03349-f006:**
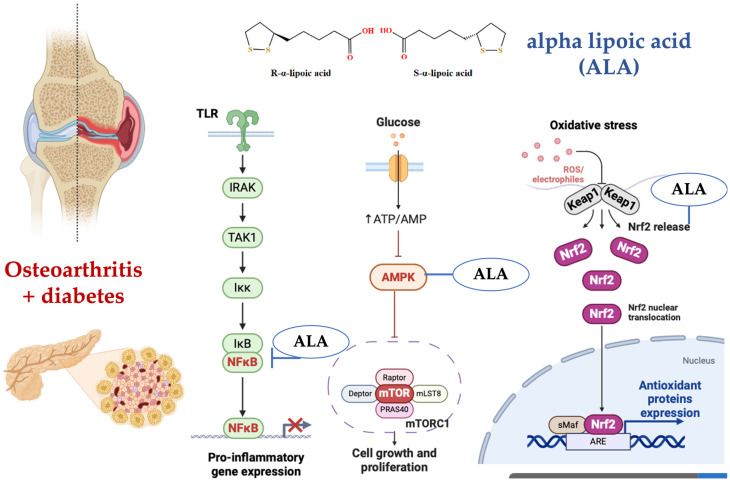
(**Left**) Alpha-lipoic acid (ALA) has been shown to inhibit the NF-κB signaling pathway in various ways. Firstly, ALA pretreatment of human large artery endothelial cells (HAEC) significantly inhibited NF-κB-binding activity induced by TNF-α in a dose-dependent manner, suggesting a metal chelation effect rather than a general antioxidation effect. Secondly, ALA has been demonstrated to block NF-κB activation, which is crucial for preventing the spread of influenza virus by inhibiting the virus-induced increase in NF-κB and caspase activities. Lastly, ALA can reduce the production of pro-inflammatory cytokines by negatively regulating NF-κB, thereby modulating the immune response and reducing inflammation. (**Center**) Alpha-lipoic acid (ALA) activates the AMPK pathway, which in turn inhibits the mTOR signaling pathway, leading to reduced cell growth and proliferation. Additionally, ALA-induced AMPK activation results in the phosphorylation and inhibition of mTOR, thereby promoting autophagy and improving cellular energy metabolism. (**Right**) The Keap1-Nrf2 pathway is a signaling pathway involved in cellular responses to oxidative stress. The regulatory protein Keap1 regulates the activity of the transcription factor Nrf2. Under conditions of oxidative stress, Nrf2 is released from Keap1 and translocates to the nucleus, where it activates the expression of genes encoding antioxidant enzymes. Alpha-lipoic acid (ALA) activates the Keap1-Nrf2 signaling pathway by modifying Keap1, leading to the release and nuclear translocation of Nrf2, which enhances the expression of antioxidant response element (ARE)-driven genes. This activation helps in reducing oxidative stress and promoting cellular defense mechanisms [84,85,86,87,88]. Figure 6 was generated using BioRender.

This study emphasizes the complex and multifaceted nature of comorbid OA and T2DM, highlighting the need for comprehensive and personalized treatment strategies. While the potential of alpha-lipoic acid as a therapeutic agent appears promising, further research is essential to fully understand its efficacy, safety, and long-term benefits in patients with both OA and T2DM.

## 5. Limitations

We acknowledge several limitations in our study. First, the relatively small sample size limits the generalizability of our findings to a wider population. A larger, multicenter study is necessary to validate these results and enhance their applicability. Additionally, the monocentric design of this study inherently restricts the population studied and may introduce selection bias. Future research should include participants from multiple centers to obtain a more representative sample. The lack of randomization in the intervention group may introduce bias into the results, highlighting a limitation of this study that should be considered in future research. Moreover, the parameters investigated were not analyzed in a cohort of patients with T2DM who do not have comorbid OA.

## 6. Conclusions

This study illustrates that the presence of type 2 diabetes mellitus in patients with osteoarthritis significantly worsens clinical symptoms and biochemical markers. Patients with both osteoarthritis and type 2 diabetes exhibited elevated levels of certain indicators of inflammation, oxidative stress, and immunological disturbances when compared to those with osteoarthritis alone. Additionally, this research identified correlations among metabolic, psychological, and inflammatory factors. This study also highlights the potential predictors of this comorbidity, including a BMI in patients with the studied comorbidity, which may serve as a potential predictor for the deterioration of the evaluated parameters.

This analysis revealed that the administration of alpha-lipoic acid in a cohort of patients with osteoarthritis and type 2 diabetes mellitus led to statistically significant improvements in WOMAC pain scores, the Lequesne Algofunctional Index, and the AIMS-P when compared to the group of patients who did not receive alpha-lipoic acid. Further research into the application and effects of alpha-lipoic acid on the progression of osteoarthritis in patients with comorbid osteoarthritis and type 2 diabetes mellitus is warranted to facilitate a personalized treatment approach.

## Figures and Tables

**Figure 1 nutrients-16-03349-f001:**
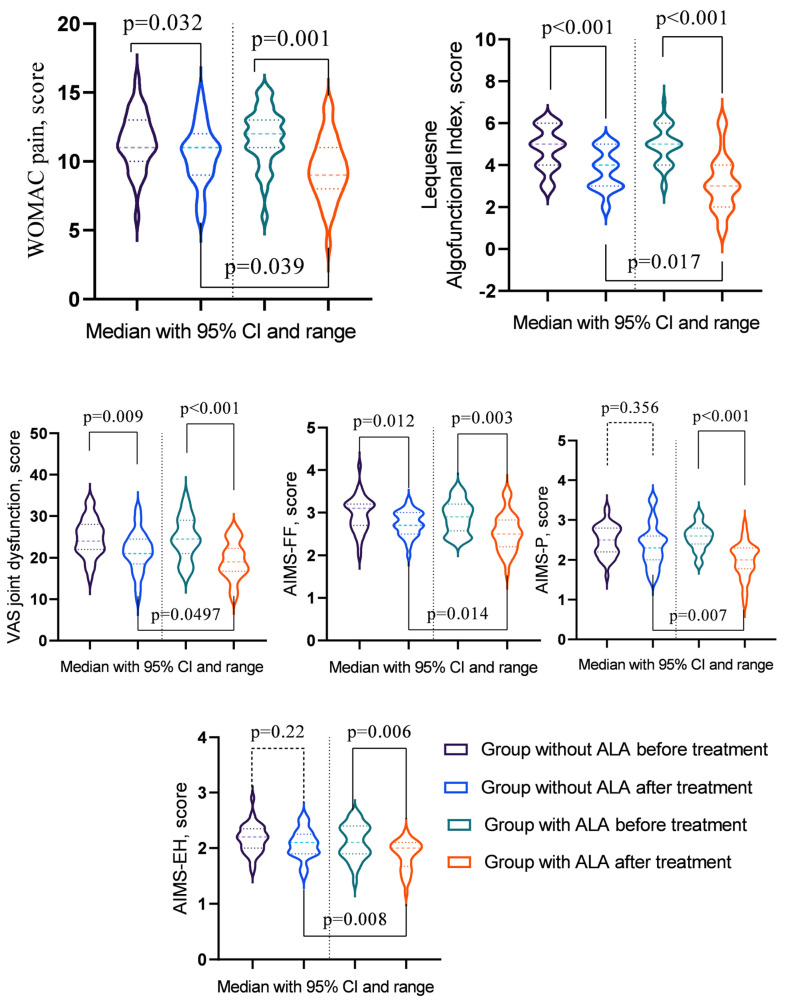
Osteoarthritis indicators showing significant changes post-treatment across groups.

**Figure 2 nutrients-16-03349-f002:**
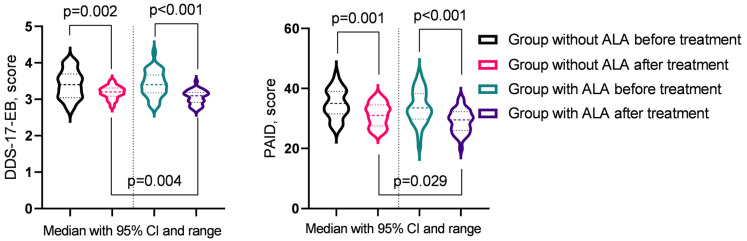
Diabetes indicators showing significant changes post-treatment across groups.

**Figure 3 nutrients-16-03349-f003:**
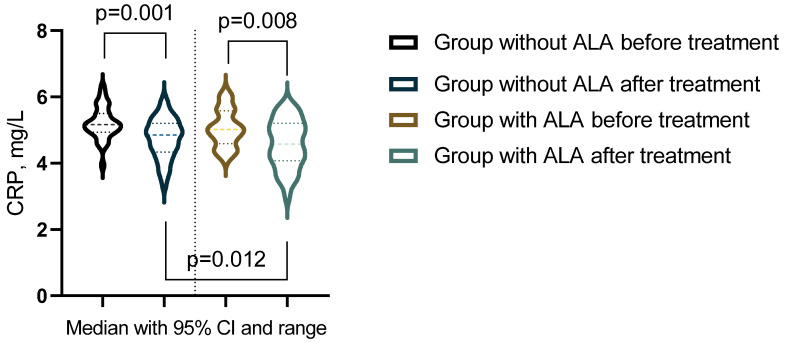
Biomarker showing significant changes post-treatment across groups.

**Figure 4 nutrients-16-03349-f004:**
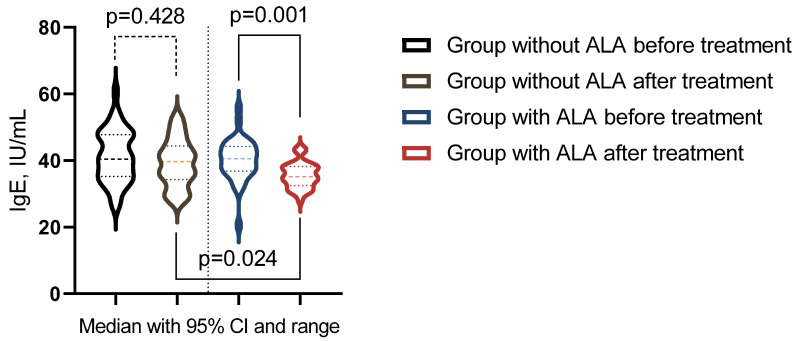
Immunological indicator showing significant changes post-treatment across groups.

**Table 1 nutrients-16-03349-t001:** Age, gender characteristics, and duration of OA in patients included in the study.

	OA (*n* = 52)	OA + T2DM (*n* = 71)	*p*-Value ^a^
Male	53.8%	54.9%	*p* = 0.7521
Age	47 (37.75–52.25)	48 (36–53.75)	*p* = 0.6143
Duration of OA	7 (5.5–9)	8 (5.0–9.5)	*p* = 0.1157

Median and interquartile range (IQR) were used to summarize the data. ^a^ Mann–Whitney.

**Table 2 nutrients-16-03349-t002:** Age, gender characteristics, and duration of OA and T2DM in patients included in the study based on the utilization of ALA in treatment.

	Group without ALA (*n* = 37)	Group with ALA (*n* = 34)	*p*-Value ^a^
Male	56.8%	52.9%	*p* = 0.3472
Age	47.5 (36.5–53)	48 (35.75–54)	*p* = 0.7981
Duration of OA	7.5 (5.0–8.5)	8 (5.25–9.75)	*p* = 0.2873
Duration of T2DM	6.5 (4.0–7.75)	6 (4.25–8)	*p* = 0.5428

Median and interquartile range (IQR) were used to summarize the data. ^a^ Mann–Whitney.

**Table 3 nutrients-16-03349-t003:** Comparison of OA indicators between patients with OA and OA + T2DM.

	OA (*n* = 52)	OA + T2DM (*n* = 71)	*p*-Value ^a^
Kellgren-Lawrence grade	2 (2–2)	2 (2–1)	*p* ^a^ = 0.469
WOMAC pain	11 (10–12)	12 (11–13)	*p* ^a^ = 0.2091
WOMAC stiffness	4 (3–5)	5 (4–5)	*p* ^a^ = 0.0003
WOMAC Physical Function	38 (36–40)	40 (36–44)	*p* ^a^ = 0.0529
Total WOMAC score	54 (51–56)	56 (53.0–60.5)	*p* ^a^ = 0.0043
Lequesne Algofunctional Index	5 (3.75–5)	5 (4–6)	*p* ^a^ = 0.0117
VAS rest pain	32 (26.75–39)	35 (32–39)	*p* ^a^ = 0.0371
VAS movement pain	52 (46–54.25)	51 (48–55)	*p* ^a^ = 0.3074
VAS inflammation	30 (28–32.25)	32 (28.5–36)	*p* ^a^ = 0.045
VAS joint dysfunction	23 (21–25)	24 (22–28)	*p* ^a^ = 0.02
AIMS-FF	2.8 (2.3–3.2)	3 (2.65–3.2)	*p* ^a^ = 0.0367
AIMS-P	2.5 (2.28–2.8)	2.6 (2.2–2.8)	*p* ^a^ = 0.9283
AIMS-SF	1.9 (1.6–2.1)	2 (1.8–2.2)	*p* ^a^ = 0.0045
AIMS-EH	2.15 (1.98–2.3)	2.2 (1.9–2.4)	*p* ^a^ = 0.5849
AIMS-GHP	2.7 (2.4–2.9)	2.9 (2.6–3.2)	*p* ^a^ = 0.0035

Median and interquartile range (IQR) were used to summarize the data. ^a^ Mann–Whitney test. WOMAC—Western Ontario and McMaster Universities Osteoarthritis Index, VAS—Visual Analog Scale, AIMS—Arthritis Impact Measurement Scales, FF—Physical Function, P—Pain, SF—Social Function, EH—Emotional Health, GHP—General Health Perception.

**Table 4 nutrients-16-03349-t004:** Diabetes-related metabolic and psychological parameter comparisons between OA and OA + T2DM patients.

	OA (*n* = 52)	OA + T2DM (*n* = 71)	*p*-Value ^a^
Fasting blood glucose, mmol/L	4.2 (4.02–4.38)	6.84 (6.51–7.63)	*p* ^a^ < 0.001
C-peptide, ng/mL	3 (2.51–3.55)	4.54 (3.81–5.14)	*p* ^a^ < 0.001
HOMA-IR	2.68 (2.47–2.83)	3.46 (3.19–3.8)	*p* ^a^ < 0.001
HbA1c, %	5.92 (5.38–6.33)	6.72 (6.55–6.86)	*p* ^a^ < 0.001
DDS-17-EB	1.2 (1.13–1.24)	3.4 (3.15–3.66)	*p* ^a^ < 0.001
DDS-17-PRD	1.1 (1.04–1.17)	4.07 (3.84–4.24)	*p* ^a^ < 0.001
DDS-17-RRD	1.13 (1.07–1.19)	4.29 (4.0–4.5)	*p* ^a^ < 0.001
DDS-17-ID	1.22 (1.13–1.26)	3.99 (3.69–4.35)	*p* ^a^ < 0.001
Total DDS-17 score	1.15 (1.13–1.19)	3.91 (3.77–4.1)	*p* ^a^ < 0.001
PAID	6 (6–6)	34 (31–39)	*p* ^a^ < 0.001

Median and interquartile range (IQR) were used to summarize the data. ^a^ Mann–Whitney test. HbA1c—Hemoglobin A1c, DDS-17—Diabetes Distress Scale-17, EB—emotional burden, PRD—physician-related distress, RRD—regimen-related distress, ID—interpersonal distress, PAID—Problem Areas in Diabetes.

**Table 5 nutrients-16-03349-t005:** Comparison of inflammatory and biochemical markers in patients with OA and OA + T2DM.

	OA (*n* = 52)	OA + T2DM (*n* = 71)	*p*-Value
Leukocytes, 10^9^/L	6.83 (6.19–7.37)	7.07 (6.64–7.57)	*p* ^a^ = 0.0664
Neutrophils, %	53.5 (52–56.25)	56 (52–61)	*p* ^a^ = 0.0332
Lymphocytes, %	21 (19–23.25)	20 (19–22)	*p* ^a^ = 0.0343
NLR	2.55 (2.33–2.7)	2.74 (2.54–3)	*p* ^a^ = 0.0003
CRP, mg/L	4.86 (4.46–5.53)	5.1 (4.83–5.51)	*p* ^a^ = 0.0624
Hydroxyproline, mg/L	1.53 (1.36–1.67)	1.61 (1.35–2.16)	*p* ^a^ = 0.066
MA, µmol/L	4.61 (4.02–5.03)	5.04 (4.54–5.42)	*p* ^a^ = 0.0008
Ceruloplasmin, mg/L	388.5 (368.2–408.5)	399 (388.5–410.2)	*p* ^a^ = 0.0088
Kallikrein, μg/L	153.45 (145–158.4)	156.8 (148.55–162.55)	*p* ^a^ = 0.0822
SOD, U/mL	53.02 (46.81–57.23)	55.3 (50.6–60.7)	*p* ^a^ = 0.0711
Catalase, U/mL	16.74 (15.84–18.24)	18.2 (16.35–20.95)	*p* ^a^ = 0.0051
α_1_-Antitrypsin, g/L	1.69 (1.63–1.74)	1.7 (1.58–1.81)	*p* ^a^ = 0.6375
α_2_-Macroglobulin, g/L	1.91 (1.86–1.99)	1.89 (1.77–2.05)	*p* ^a^ = 0.3204

Median and interquartile range (IQR) were used to summarize the data. ^a^ Mann–Whitney test. NLR—neutrophil-to-lymphocyte ratio, CRP—C-Reactive Protein, MA—malonaldehyde, SOD—superoxide dismutase.

**Table 6 nutrients-16-03349-t006:** Comparison of immunological markers in patients with OA and OA + T2DM.

	OA (*n* = 52)	OA + T2DM (*n* = 71)	*p*-Value
Serum IgA, g/L	1.84 (1.73–1.97)	1.93 (1.78–2.11)	*p* ^a^ = 0.0218
IgM, g/L	1.02 (0.9–1.11)	1.05 (0.94–1.21)	*p* ^a^ = 0.1328
IgG, g/L	9.12 (8.49–9.66)	9.08 (8.25–9.68)	*p* ^a^ = 0.6616
IgE, IU/mL	38.8 (37.18–39.59)	40.4 (36.3–45.85)	*p* ^a^ = 0.0214
T-Lymphocytes (CD3+, CD19−), %	59.9 (56.78–62.75)	60.6 (57.5–65.55)	*p* ^a^ = 0.0896
T-Helpers (CD4+, CD8−), %	42.55 (40.35–43.43)	43.5 (40.8–47.4)	*p* ^a^ = 0.0795
T-Cytotoxic Cells (CD4−, CD8+), %	28 (25.48–30.65)	27.7 (26.35–29.75)	*p* ^a^ = 0.7645
Immunoregulatory Index	1.5 (1.4–1.6)	1.6 (1.41–1.72)	*p* ^a^ = 0.047
Cytotoxic Cells (CD3+, CD56+), %	4.7 (4.4–5.03)	4.9 (4.3–5.45)	*p* ^a^ = 0.2383
NK Cells (CD3−, CD56+), %	9.5 (9.2–9.8)	9.7 (8.8–10.65)	*p* ^a^ = 0.2447
B-Lymphocytes (CD3−, CD19+), %	9.75 (9.3–10.3)	10.1 (9.6–10.8)	*p* ^a^ = 0.0709
Monocytes/Macrophages (CD14), %	8.1 (7.9–8.4)	8.2 (7.5–8.85)	*p* ^a^ = 0.5743

Median and interquartile range (IQR) were used to summarize the data. ^a^ Mann–Whitney test.

**Table 7 nutrients-16-03349-t007:** Comparison of osteoarthritis indicators before and after treatment in groups with and without ALA.

	Group without ALA (*n* = 37)		Group with ALA (*n* = 34)		*p*-Value ^bc^
	Before Treatment	After Treatment	Before Treatment	After Treatment	
Kellgren-Lawrence grade	2 (1–2)	2 (1–2)	2 (1.25–2)	2 (1–2)	*p* ^b^ = 0.7395
*p*-value ^ac^	*p*^a^ = 0.233		*p*^a^ = 0.484	*p*^d^ = 9813	*p* ^c^ = 0.7298
WOMAC pain	11 (10–13)	11 (9–12)	12 (11–13)	9 (8–11)	*p* ^b^ = 0.4518
*p*-value ^ac^	*p* ^a^ = 0.0321		*p* ^a^ = 0.0014	*p* ^d^ = 0.0364	*p* ^c^ = 0.0390
WOMAC stiffness	5 (4–6)	4 (4–5)	5 (4–5)	4 (3–4)	*p* ^b^ = 0.1993
*p*-value ^ac^	*p* ^a^ = 0.0010		*p* ^a^ = 0.0055	*p* ^d^ = 0.8481	*p* ^c^ = 0.1937
WOMAC Physical Function	40 (35–44)	37 (35–39)	40 (36.25–43.75)	36.5 (35–38.75)	*p* ^b^ = 0.7996
*p*-value ^ac^	*p* ^a^ = 0.055		*p* ^a^ = 0.0049	*p* ^d^ = 0.7032	*p* ^c^ = 0.9538
Total WOMAC score	56 (53–61)	52 (48–53)	56.5 (53.25–59.75)	50 (47–53)	*p* ^b^ = 0.8853
*p*-value ^ac^	*p* ^a^ = 0.0028		*p* ^a^ = < 0.001	*p* ^d^ = 0.3222	*p* ^c^ = 0.1876
Lequesne Algofunctional Index	5 (4–6)	4 (3–5)	5 (4–5.75)	3 (2–4)	*p* ^b^ = 0.5299
*p*-value ^ac^	*p* ^a^ = 0.0006		*p* ^a^ = < 0.001	*p* ^d^ = 0.0232	*p* ^c^ = 0.0172
VAS rest pain	37 (32–40)	30 (27–34)	34 (32–38)	30.5 (28–34)	*p* ^b^ = 0.1541
*p* ^ac^-value	*p* ^a^ < 0.001		*p* ^a^ < 0.001	*p* ^d^ = 0.3632	*p* ^c^ = 0.751
VAS movement pain	52 (48–54)	50 (43–53)	50.5 (46.25–58.5)	46.5 (39.5–50.75)	*p* ^b^ = 0.624
*p*-value ^ac^	*p* ^a^ = 0.0461		*p* ^a^ = 0.0049	*p* ^d^ = 0.2269	*p* ^c^ = 0.0619
VAS inflammation	32 (28–34)	27 (22–31)	33 (29.25–37)	26 (24–28.75)	*p* ^b^ = 0.3127
*p*-value ^ac^	*p* ^a^ = 0.0113		*p* ^a^ < 0.001	*p* ^d^ = 0.2819	*p* ^c^ = 0.5757
VAS joint dysfunction	24 (22–28)	21 (19–24)	24.5 (21.25–28.75)	19 (17–22)	*p* ^b^ = 0.9586
*p*-value ^ac^	*p* ^a^ = 0.0089		*p* ^a^ = 0.0004	*p* ^d^ = 0.1850	*p* ^c^ = 0.0497
AIMS-FF	3.1 (2.7–3.2)	2.7 (2.5–3)	2.9 (2.6–3.18)	2.5 (2.23–2.8)	*p* ^b^ = 0.3611
*p*-value ^ac^	*p* ^a^ = 0.0121		*p* ^a^ = 0.0026	*p* ^d^ = 0.2024	*p* ^c^ = 0.0142
AIMS-P	2.5 (2.2–2.8)	2.3 (2–2.6)	2.6 (2.4–2.78)	2 (1.8–2.3)	*p* ^b^ = 0.4018
*p*-value ^ac^	*p* ^a^ = 0.3564		*p* ^a^ < 0.001	*p* ^d^ = 0.0018	*p* ^c^ = 0.0073
AIMS-SF	2 (1.9–2.2)	1.9 (1.7–2)	2 (1.73–2.2)	1.75 (1.6–1.9)	*p* ^b^ = 0.3663
*p*-value ^ac^	*p* ^a^ = 0.0241		*p* ^a^ = 0.0146	*p* ^d^ = 0.6271	*p* ^c^ = 0.1003
AIMS-EH	2.2 (2–2.3)	2.1 (1.9–2.2)	2.1 (1.9–2.4)	2 (1.7–2.1)	*p* ^b^ = 0.5388
*p*-value ^ac^	*p* ^a^ = 0.2196		*p* ^a^ = 0.0055	*p* ^d^ = 0.1107	*p* ^c^ = 0.0083
AIMS-GHP	2.9 (2.6–3.1)	2.7 (2.4–3)	2.9 (2.7–3.28)	2.6 (2.43–2.7)	*p* ^b^ = 0.3616
*p*-value ^ac^	*p* ^a^ = 0.0907		*p* ^a^ = 0.0004	*p* ^d^ = 0.1125	*p* ^c^ = 0.2377

Median and interquartile range (IQR) were used to summarize the data. ^a^ The statistical difference observed within the group before and after treatment, based on the Wilcoxon signed-rank test. ^b^ The statistical difference between groups before treatment, based on the Mann–Whitney U test. ^c^ The statistical difference between groups after treatment, based on the Mann–Whitney U test. ^d^ The statistical difference of the Difference-in-Differences test.

**Table 8 nutrients-16-03349-t008:** Comparison of diabetes indicators before and after treatment in groups with and without ALA.

	Group without ALA (*n* = 37)		Group with ALA (*n* = 34)		*p*-Value ^bc^
	Before Treatment	After Treatment	Before Treatment	After Treatment	
Fasting blood glucose, mmol/L	6.87 (6.5–7.24)	6.52 (6.17–6.8)	6.8 (6.54–7.79)	6.47 (5.8–6.99)	*p* ^b^ = 0.427
*p*-value ^ac^	*p* ^a^ = 0.0271		*p* ^a^ = 0.003	*p* ^d^ = 0.3950	*p* ^c^ = 0.881
C-peptide, ng/mL	4.49 (3.88–4.96)	4.23 (3.78–4.92)	4.55 (3.56–5.58)	4.22 (3.88–4.79)	*p* ^b^ = 0.8674
*p*-value ^ac^	*p* ^a^ = 0.4689		*p* ^a^ = 0.1768	*p* ^d^ = 5887	*p* ^c^ = 0.8539
HOMA-IR	3.49 (3.19–3.75)	3.4 (3.18–3.6)	3.45 (3.2–3.83)	3.35 (2.87–3.69)	*p* ^b^ = 0.8992
*p*-value ^ac^	*p* ^a^ = 0.261		*p* ^a^ = 0.093	*p* ^d^ = 0.5498	*p* ^c^ = 0.6452
HbA1c, %	6.7 (6.38–7.25)	6.49 (6.34–6.96)	6.73 (6.63–6.79)	6.63 (6.41–6.94)	*p* ^b^ = 0.9449
*p*-value ^ac^	*p* ^a^ = 0.2642		*p* ^a^ = 0.3308	*p* ^d^ = 6410	*p* ^c^ = 0.5805
DDS-17-EB	3.4 (3.05–3.66)	3.2 (3.11–3.29)	3.4 (3.18–3.65)	3.1 (2.94–3.19)	*p* ^b^ = 0.9266
*p*-value ^ac^	*p* ^a^ = 0.0022		*p* ^a^ = 0.0001	*p* ^d^ = 0.1603	*p* ^c^ = 0.0035
DDS-17-PRD	4.07 (3.82–4.23)	3.89 (3.51–4.11)	4.11 (3.92–4.25)	3.73 (3.42–4.02)	*p* ^b^ = 0.4439
*p*-value ^ac^	*p* ^a^ = 0.1843		*p* ^a^ = 0.0022	*p* ^d^ = 3343	*p* ^c^ = 0.4406
DDS-17-RRD	4.3 (4.03–4.53)	3.88 (3.69–4.18)	4.25 (3.99–4.47)	3.85 (3.72–3.98)	*p* ^b^ = 0.7385
*p*-value ^ac^	*p* ^a^ = 0.0036		*p* ^a^ < 0.001	*p* ^d^ = 0.8931	*p* ^c^ = 0.4718
DDS-17-ID	3.93 (3.57–4.29)	3.86 (3.41–4.2)	4.02 (3.75–4.41)	3.66 (3.3–3.96)	*p* ^b^ = 0.4007
*p*-value ^ac^	*p* ^a^ = 0.3898		*p* ^a^ = 0.0183	*p* ^d^ = 0.3343	p^c^ = 0.4405
Total DDS-17 score	3.88 (3.76–4.1)	3.71 (3.57–3.78)	3.97 (3.77–4.09)	3.55 (3.45–3.72)	*p* ^b^ = 0.9634
*p*-value ^ac^	*p* ^a^ = 0.6635		*p* ^a^ = 0.7301	*p* ^d^ = 0.0591	*p* ^c^ = 0.7596
PAID	35 (32–39)	31 (28–34)	33.5 (30.25–37.75)	29.5 (26.25–31.75)	*p* ^b^ = 0.3837
*p*-value ^ac^	*p* ^a^ = 0.0012		*p* ^a^ = 0.0002	*p* ^d^ = 0.5524	*p* ^c^ = 0.0293

Median and interquartile range (IQR) were used to summarize the data. ^a^ The statistical difference observed within the group before and after treatment, based on the Wilcoxon signed-rank test. ^b^ The statistical difference between groups before treatment, based on the Mann–Whitney U test. ^c^ The statistical difference between groups after treatment, based on the Mann–Whitney U test. ^d^ The statistical difference of the Difference-in-Differences test.

**Table 9 nutrients-16-03349-t009:** Comparison of biomarkers before and after treatment in groups with and without ALA.

	Group without ALA (*n* = 37)		Group with ALA (*n* = 34)		*p*-Value ^bc^
	Before Treatment	After Treatment	Before Treatment	After Treatment	
Leukocytes, 10^9^/L	7.26 (6.64–7.63)	7.04 (6.19–7.43)	7.02 (6.63–7.29)	6.55 (6.06–6.87)	*p* ^b^ = 0.2224
*p*-value ^ac^	*p* ^a^ = 0.1534		*p* ^a^ = 0.0024	*p* ^d^ = 0.4177	*p* ^c^ = 0.0622
Neutrophils, %	56 (50–59)	54 (50–56)	57 (53–61,75)	55 (50.25–61.75)	*p* ^b^ = 0.0889
*p*-value	*p* ^a^ = 0.1366		*p* ^a^ = 0.0054	*p* ^d^ = 0.2945	*p* ^c^ = 0.2517
Lymphocytes, %	19 (19–21)	21 (19–22)	20 (19–23.75)	22 (20–24)	*p* ^b^ = 0.1237
*p*-value ^ac^	*p* ^a^ = 0.3266		*p* ^a^ = 0.1269	*p* ^d^ = 0.3988	*p* ^c^ = 0.0078
NLR	2.74 (2.55–2.94)	2.52 (2.39–2.78)	2.76 (2.54–3.08)	2.42 (2.25–2.68)	*p* ^b^ = 0.4789
*p*-value ^ac^	*p* ^a^ = 0.0132		*p* ^a^ = 0.0023	*p* ^d^ = 0.1228	*p* ^c^ = 0.0893
CRP, mg/L	5.16 (4.95–5.48)	4.85 (4.38–5.2)	5.02 (4.63–5.54)	4.58 (4.11–5.16)	*p* ^b^ = 0.1286
*p*-value ^ac^	*p* ^a^ = 0.001		*p* ^a^ = 0.0078	*p* ^d^ = 0.7379	*p* ^c^ = 0.0119
Hydroxyproline, mg/L	1.61 (1.22–1.87)	1.64 (1.32–1.8)	1.64 (1.38–2.27)	1.47 (1.12–2.04)	*p* ^b^ = 0.497
*p*-value ^ac^	*p* ^a^ = 0.7571		*p* ^a^ = 0.1324	*p* ^d^ = 0.6658	*p* ^c^ = 0.7472
MA, µmol/L.	5.04 (4.69–5.39)	4.81 (4.32–5.42)	5.04 (4.44–5.6)	4.44 (3.93–5.05)	*p* ^b^ = 0.8449
*p*-value ^ac^	*p* ^a^ = 0.1056		*p* ^a^ = 0.0031	*p* ^d^ = 0.2647	*p* ^c^ = 0.0546
Ceruloplasmin, mg/L	397.5 (387.1–410.5)	390.2 (383.5–409.3)	405 (391–408.4)	388.1 (372.3–397.6)	*p* ^b^ = 0.5118
*p*-value ^ac^	*p* ^a^ = 0.1491		*p* ^a^ = 0.0019	*p* ^d^ = 0.1113	*p* ^c^ = 0.1272
Kallikrein, μg/L	153.7 (148.2–162.4)	152.1 (146.7–156.9)	157.5 (150.4–162.5)	152.2 (145–159.5)	*p* ^b^ = 0.5885
*p*-value ^ac^	*p* ^a^ = 0.2364		*p* ^a^ = 0.1043	*p* ^d^ = 0.4760	*p* ^c^ = 0.7912
SOD, U/mL	57.5 (52.2–61.5)	59.3 (55.5–63.3)	54.75 (47.83–59.68)	58.3 (53.7–63.58)	*p* ^b^ = 0.1837
*p*-value ^ac^	*p* ^a^ = 0.2189		*p* ^a^ = 0.0033	*p* ^d^ = 0.5047	*p* ^c^ = 0.5229
Catalase, U/mL	18.3 (16.8–20.1)	20.5 (18.43–23.26)	17.7 (15.23–21.73)	21.95 (20.45–23.98)	*p* ^b^ = 0.8314
*p*-value ^ac^	*p* ^a^ = 0.0146		*p* ^a^ = 0.0014	*p* ^d^ = 0.3173	*p* ^c^ = 0.1258
α_1_-Antitrypsin, g/L	1.7 (1.57–1.78)	1.7 (1.51–1.76)	1.71 (1.59–1.83)	1.61 (1.45–1.79)	*p* ^b^ = 0.6083
*p*-value ^ac^	*p* ^a^ = 0.7571		*p* ^a^ = 0.1744	*p* ^d^ = 0.4584	*p* ^c^ = 0.4303
α_2_-Macroglobulin, g/L	1.88 (1.78–1.99)	1.99 (1.79–2.09)	1.9 (1.7–2.11)	1.93 (1.84–2.12)	*p* ^b^ = 0.7956
*p*-value ^ac^	*p* ^a^ = 0.245		*p* ^a^ = 0.0889	*p* ^d^ = 0.5270	*p* ^c^ = 0.6827

Median and interquartile range (IQR) were used to summarize the data. ^a^ The statistical difference observed within the group before and after treatment, based on the Wilcoxon signed-rank test. ^b^ The statistical difference between groups before treatment, based on the Mann–Whitney U test. ^c^ The statistical difference between groups after treatment, based on the Mann–Whitney U test. ^d^ The statistical difference of the Difference-in-Differences test.

**Table 10 nutrients-16-03349-t010:** Comparison of immunological markers before and after treatment in groups with and without ALA.

	Group without ALA (*n* = 37)		Group with ALA (*n* = 34)		*p*-Value ^bc^
	Before Treatment	After Treatment	Before Treatment	After Treatment	
Serum IgA, g/L	1.93 (1.79–2.11)	1.87 (1.69–2.08)	1.92 (1.76–2.09)	1.84 (1.56–2.04)	*p* ^b^ = 0.9175
*p*-value ^ac^	*p* ^a^ = 0.02204		*p* ^a^ = 0.0251	*p* ^d^ = 0.5421	*p* ^c^ = 0.3422
IgM, g/L	1.08 (1.02–1.21)	0.99 (0.84–1.19)	0.99 (0.89–1.2)	0.99 (0.71–1.13)	*p* ^b^ = 0.0811
*p*-value ^ac^	*p* ^a^ = 0.2133		*p* ^a^ = 0.1438	*p* ^d^ = 0.8615	*p* ^c^ = 0.4071
IgG, g/L	8.74 (8.26–9.41)	8.81 (8.23–9.76)	9.22 (8.19–9.74)	8.59 (8.17–9.45)	*p* ^b^ = 0.4204
*p*-value ^ac^	*p* ^a^ = 0.7399		*p* ^a^ = 0.3303	*p* ^d^ = 0.2929	*p* ^c^ = 0.6786
IgE, IU/mL	40.4 (35.4–47.6)	39.7 (34.6–43.9)	40.6 (36.93–44.15)	35.15 (32.75–38.08)	*p* ^b^ = 0.704
*p*-value ^ac^	*p* ^a^ = 0.4275		*p* ^a^ = 0.0008	*p* ^d^ = 0.1145	*p* ^c^ = 0.0244
T-Lymphocytes (CD3+, CD19−), %	61 (56.6–65.4)	57.2 (53.8–64.9)	60 (57.58–66.03)	58.6 (53.13–63.18)	*p* ^b^ = 0.7255
*p*-value ^ac^	*p* ^a^ = 0.1017		*p* ^a^ = 0.099	*p* ^d^= 0.8525	*p* ^c^ = 0.8001
T-Helpers (CD4+, CD8−), %	43.5 (41–46.9)	42.6 (40.1–44.9)	44.95 (40.28–47.85)	40.45 (38.7–44.98)	*p* ^b^ = 0.872
*p*-value ^ac^	*p* ^a^ = 0.4736		*p* ^a^ = 0.0566	*p* ^d^ = 0.2574	*p* ^c^ = 0.2617
T-Cytotoxic Cells (CD4−, CD8+), %	28.2 (26.4–29.8)	29.5 (26–30.8)	27.45 (26.08–29.5)	28.45 (27.1–32.25)	*p* ^b^ = 0.508
*p*-value ^ac^	*p* ^a^ = 0.6838		*p* ^a^ = 0.513	*p* ^d^ = 0.1782	*p* ^c^ = 0.2766
Immunoregulatory Index	1.61 (1.39–1.64)	1.47 (1.4–1.67)	1.59 (1.42–1.76)	1.4 (1.27–1.55)	*p* ^b^ = 0.6247
*p*-value ^ac^	*p* ^a^ = 0.2677		*p* ^a^ = 0.0125	*p* ^d^ = 0.0580	*p* ^c^ = 0.056
Cytotoxic Cells (CD3+, CD56+), %	4.8 (4.3–5.5)	4.8 (4.4–5.2)	4.9 (4.33–5.4)	4.85 (4.3–5.1)	*p* ^b^ = 0.7294
*p*-value ^ac^	*p* ^a^ = 0.5724		*p* ^a^ = 0.6939	*p* ^d^ = 0.9795	*p* ^c^ = 0.7251
NK Cells (CD3−, CD56+), %	10.3 (9–10.8)	9.5 (8.7–10.7)	9.4 (8.73–10.55)	9.15 (8.5–9.78)	*p* ^b^ = 0.2332
*p*-value ^ac^	*p* ^a^ = 0.1996		*p* ^a^ = 0.3879	*p* ^d^ = 0.5497	*p* ^c^ = 0.1389
B-Lymphocytes (CD3−, CD19+), %	10 (9.7–10.5)	9.9 (9.3–10.7)	10.2 (9.45–10.9)	9.65 (8.03–10.45)	*p* ^b^ = 0.7427
*p*-value ^ac^	*p* ^a^ = 0.8741		*p* ^a^ = 0.0887	*p* ^d^ = 0.1601	*p* ^c^ = 0.1499
Monocytes/Macrophages (CD14), %	8.2 (7.5–9)	8.1 (7.2–8.7)	8.15 (7.55–8.58)	7.65 (7.13–8.48)	*p* ^b^ = 0.5684
*p*-value ^ac^	*p* ^a^ = 0.2233		*p* ^a^ = 0.2814	*p* ^d^ = 0.9475	*p* ^c^ = 0.4403

Median and interquartile range (IQR) were used to summarize the data. ^a^ The statistical difference observed within the group before and after treatment, based on the Wilcoxon signed-rank test. ^b^ The statistical difference between groups before treatment, based on the Mann–Whitney U test. ^c^ The statistical difference between groups after treatment, based on the Mann–Whitney U test. ^d^ The statistical difference of the Difference-in-Differences test.

## Data Availability

The original contributions presented in the study are included in the article, further inquiries can be directed to the corresponding authors.

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
