# Peer review of "Exploring the Efficacy of Alpha-Lipoic Acid in Comorbid Osteoarthritis and Type 2 Diabetes Mellitus"

_nutrients, 2024, doi:10.3390/nu16193349_

Round 1
Reviewer 1 Report
Comments and Suggestions for Authors
This is an interesting study showing first significant differences in pain, inflammation and other features between OA patients with and without T2D and then showing the beneficial effects on pain and pain related traits in OA patients resulting from supplementation with ALA.
Although the findings are potentially interesting there are major aspects not properly addressed in the study.
First, this appear to be two studies merged together as the direct relevance of the first study to the second is not clear. One study is about the differences between OA patients with or without T2D and the BMI of participants is not included and none of the analyses are adjusted for BMI. BMI is well known to influence pain severity of OA therefore the analyses in Tables 3,4,5 and 6 need to be adjusted for BMI. If BMI is higher in individuals with T2D that might mediate all the differences observed and this needs to be addressed. This section of the paper contains a very high number of figures and tables which can be merged and shown in a more concise manner. Importantly this very long section of the paper doesn't really have a nutritional component.
The second part of the paper, is the actual nutritional component of the study which shows longitudinal changes in various OA-related traits in a group receiving ALA supplementation and a control group receiving standard care. A CONSORT diagram and information on participant BMI is missing. There isn't a single primary outcome and the authors have tested a large number of traits therefore the study requires adjustment for multiple tests. Moreover, improvements in pain traits were seen in the group not receiving ALA making interpretation of the results very challenging.
The conclusions are not really supported by the findings unless the authors can show a mediatory effect of the traits that change with ALA but not without ALA on the pain and functional traits.
Author Response
We thank the reviewer for the evaluation and constructive feedback.
Q1. First, this appear to be two studies merged together as the direct relevance of the first study to the second is not clear
A1. In the first part of the study, we investigated the impact of comorbidity between osteoarthritis and diabetes on the studied parameters. As our analysis revealed a statistically significant worsening in some of the evaluated outcomes, including the progression of osteoarthritis, diabetes, inflammation, oxidative stress, and immunogram indicators in a group of patients with comorbid osteoarthritis and type 2 diabetes, we decided to further explore the effect of alpha-lipoic acid on these parameters in this cohort of patients with comorbid osteoarthritis and type 2 diabetes. The cohort was then divided into two subgroups: one that received alpha-lipoic acid treatment and another that did not, allowing us to assess the differential impact of the intervention more precisely.
Q2. One study is about the differences between OA patients with or without T2D and the BMI of participants is not included and none of the analyses are adjusted for BMI. BMI is well known to influence pain severity of OA therefore the analyses in Tables 3,4,5 and 6 need to be adjusted for BMI. If BMI is higher in individuals with T2D that might mediate all the differences observed and this needs to be addressed.
A2. The authors acknowledge that BMI can significantly influence the severity of pain in osteoarthritis. The level of BMI was included into our study and a statistically significant difference was established. It may serve as a potential predictor of the outcomes under conditions of comorbidity between osteoarthritis and type 2 diabetes. We will ensure that future studies are adjusteed for BMI in the analyses.
Q3. This section of the paper contains a very high number of figures and tables which can be merged and shown in a more concise manner. Importantly this very long section of the paper doesn't really have a nutritional component.
A3. Based on this comment, we have now removed Figures 1, 2, 4, and 5 to enhance the clarity and conciseness of the paper. We believe this revision will improve the overall readability and focus of the study.
Q4. The second part of the paper, is the actual nutritional component of the study which shows longitudinal changes in various OA-related traits in a group receiving ALA supplementation and a control group receiving standard care. A CONSORT diagram and information on participant BMI is missing. There isn't a single primary outcome and the authors have tested a large number of traits therefore the study requires adjustment for multiple tests. Moreover, improvements in pain traits were seen in the group not receiving ALA making interpretation of the results very challenging.
A4. We have now revised the manuscript and included the CONSORT diagram in the Materials and Methods section to clarify the randomization of participants in the study. Additionally, we have now added a Difference-in-Differences (DiD) statistical analysis to better assess the statistical significance of the obtained results. We have also incorporated BMI, as it plays a significant role in the studied parameters.
Q5. The conclusions are not really supported by the findings unless the authors can show a mediatory effect of the traits that change with ALA but not without ALA on the pain and functional traits.
A5. We have now revised the conclusions following this recommendations, and we have incorporated indicators that demonstrated statistically significant changes with the administration of alpha-lipoic acid in patients with comorbid osteoarthritis and type 2 diabetes mellitus.
We thank again the reviewer for the feedback
Reviewer 2 Report
Comments and Suggestions for Authors
The manuscript entitled Exploring the Efficacy of Alpha-Lipoic Acid in Comorbid Osteoarthritis and Type 2 Diabetes Mellitus is an original paper. The authors utilized various assessment methods to measure the patient cohorts' inflammation, oxidative stress, and glycemic control to analyze the exacerbation of clinical symptoms and biochemical markers in patients with osteoarthritis and type 2 diabetes mellitus compared to those with osteoarthritis. The patients with comorbid osteoarthritis and type 2 diabetes mellitus received alpha-lipoic acid. It seems that the treatment with alpha-lipoic acid resulted in improved joint function, reduced inflammation and oxidative stress levels, and better glycemic control. This study has a small number of patients with many and diverse parameters, which make it difficult to understand from the novelty point of view. In my opinion, this study is just an analysis of a large data, in one hospital, in a small group of patients, without any new data important for clinical practice.
Major revision
On page 2 the authors have said: ``Studies show that ALA significantly benefits the management of osteoarthritis in patients with DM.`` and, also, on page 21: `` These findings are consistent with previous studies that have reported similar associations between T2DM and the worsening of OA.`` According with these statements, what are the new data offered by this study? The comments on page 21 are not convincing, unfortunately.
How do you explain that SOD (neither CRP) was not statistically significant in table 5 even NLR was significantly?
All correlations index in figure 1 and 2 are weak and very weak. Actually, there are not strong positive correlation in figure 1 and 2. In addition, in statistics correlation in not causality. Therefore, caution is necessary when the authors interpret these data.
Page 22 lines 518-522: What do you think that are the others factors which contribute to enhancing glycemic control in your study? It is important underlying this.
Minor revision
In table 1 is no need including both gender; usually men is sufficient. The same comment for table 2.
Please explain all acronyms in tables.
Author Response
We thank the reviewer for evaluating the manuscript and constructive feedback.
Q1. On page 2 the authors have said: ``Studies show that ALA significantly benefits the management of osteoarthritis in patients with DM.`` and, also, on page 21: `` These findings are consistent with previous studies that have reported similar associations between T2DM and the worsening of OA.`` According with these statements, what are the new data offered by this study? The comments on page 21 are not convincing, unfortunately.
A1. The current study complements previous research on the comorbidity of osteoarthritis and type 2 diabetes mellitus by examining the impact of this comorbidity on the progression of osteoarthritis and providing new insights into the effects of alpha-lipoic acid administration. We have enhanced our analysis with a difference-in-differences approach, and the conclusions have been revised to include only those indicators that experienced statistically significant changes in accordance with our findings. Additionally, we included body mass index (BMI) as a potential predictor of the deterioration of the evaluated parameters in the context of comorbidity between osteoarthritis and type 2 diabetes mellitus.
Q2. How do you explain that SOD (neither CRP) was not statistically significant in table 5 even NLR was significantly?
A2. The lack of statistical significance for SOD and CRP in Table 5, despite NLR being significant, may be explained by differences in the underlying biological processes and the nature of the inflammatory responses they represent. NLR (Neutrophil-to-Lymphocyte Ratio) is a composite marker that reflects both innate and adaptive immune responses, which may make it a more sensitive indicator of systemic inflammation in the context of our study. On the other hand, SOD (Superoxide Dismutase) is an antioxidant enzyme, and its levels may not fluctuate in direct correlation with acute inflammatory markers or may be influenced by other compensatory mechanisms. Similarly, CRP (C-Reactive Protein) is a more general marker of inflammation and might not capture the specific inflammatory processes that NLR is more sensitive to in this particular context.
Q3. All correlations index in figure 1 and 2 are weak and very weak. Actually, there are not strong positive correlation in figure 1 and 2. In addition, in statistics correlation in not causality. Therefore, caution is necessary when the authors interpret these data.
A3. In light of this feedback, we have removed Figures 1 and 2 from the study to enhance clarity and streamline the article. Furthermore, we have reduced references to correlation in the conclusions to promote a more cautious interpretation of the findings.
Q4. Page 22 lines 518-522: What do you think that are the others factors which contribute to enhancing glycemic control in your study? It is important underlying this.
A4. We have now revised and expanded the information in the article to include additional factors that may contribute to enhancing glycemic control in our study.
Q5. In table 1 is no need including both gender; usually men is sufficient. The same comment for table 2.
A5. We have now made the necessary adjustments to Tables 1 and 2 in accordance with this recommendation, including only men where appropriate.
Q6. Please explain all acronyms in tables.
A6. We have now provided explanations for all acronyms below the tables to enhance clarity.
We thank again the reviewer for the feedback
Reviewer 3 Report
Comments and Suggestions for Authors
Authors used an intervention study to examine an effect of Alpha-Lipoic Acid on indicators of osteoarthritis and diabetes and biomarkers among patients with osteoarthritis and diabetes. However, this article has not fully answered some of the questions due to insufficient description.
First, authors suggest “Inclusion criteria for the study encompassed individuals of both genders who had a confirmed diagnosis of hip and knee osteoarthritis (ICD-10 codes M16 and M17) and type 2 diabetes mellitus (ICD-10 code E11).” (L121), but they do not explain the details of how to diagnoses hip and knee osteoarthritis and type 2 diabetes mellitus (e.g., levels of serum glucose and HbA1c). Without the detailed descriptions, it is difficult for readers to understand what they did. Authors should add detailed descriptions regarding how to diagnoses hip and knee osteoarthritis and type 2 diabetes mellitus.
Second, authors showed odds of OA+T2DM for HOMA-IR, HbA1C, and serum IgA (table 7 and Figure 3), but it is difficult to understand why they showed this analysis. Authors may calculate propensity scores, but they did not use them in main analyses (i.e., table 8-11). Moreover, it is difficult to understand what they did in table 7 and figure 3 (e.g., what is “group” in title of table 7? and what is population authors used in these analyses? What did authors adjust for?). Authors should explain why they showed these analyses and what they did in this manuscript.
Third, authors showed intervention using Alpha-Lipoic Acid, but they do not show difference of difference. Authors showed “statistical difference between groups before treatment” and “statistical difference between groups after treatment,”, but these results may be biased due to small sample size and skewed distributions. Authors should add results regarding difference of difference in table 8-11.
Fourth, authors do not randomize the intervention group, which may lead to biased results. Authors should add limitation regarding his issue.
Finally, authors described some of sentences without citation or justification as follows; “OA is a degenerative joint disorder that primarily targets the articular cartilage and is influenced by factors such as trauma, metabolic processes, and comorbidities.” (P44), “standard protocols” (P114), “Alpha lipoic acid (ALA) reduces the formation of reactive oxygen species (ROS) by directly scavenging free radicals, including hydroxyl radicals and hydrogen peroxide. Additionally, ALA chelates transition metal ions, preventing the conversion of weak oxidants into more harmful reactive species. ALA also enhances the levels of intracellular antioxidants such as glutathione, which further helps in neutralizing ROS. Moreover, ALA can regenerate other antioxidants like vitamin C and vitamin E, thereby maintaining a robust antioxidant defense system within the cell. ” (P511), “Alpha lipoic acid (ALA) has been shown to inhibit the NF-κB signaling pathway in various ways. Firstly, ALA pretreatment of human large artery endothelial cells (HAEC) significantly inhibited NF-κB binding activity induced by TNF-α in a dose-dependent manner, suggesting a metal chelation effect rather than a general antioxidation effect. Secondly, ALA has been demonstrated to block NF-κB activation, which is crucial for preventing the spread of influenza virus by inhibiting the virus-induced increase in NF-κB and caspase activities. Lastly, ALA can reduce the production of pro-inflammatory cytokines by negatively regulating NF-κB, thereby modulating the immune response and reducing inflammation.(center) Alpha lipoic acid (ALA) activates the AMPK pathway, which in turn inhibits the mTOR signaling pathway, leading to reduced cell growth and proliferation. Additionally, ALA-induced AMPK activation results in the phosphorylation and inhibition of mTOR, thereby promoting autophagy and improving cellular energy metabolism. (right) The Keap1-Nrf2 pathway is a signaling pathway involved in cellular responses to oxidative stress. The regulatory protein Keap1 regulates the activity of the transcription factor Nrf2. Under conditions of oxidative stress, Nrf2 is released from Keap1 and translocates to the nucleus, where it activates the expression of genes encoding antioxidant enzymes. Alpha lipoic acid (ALA) activates the Keap1-Nrf2 signaling pathway by modifying Keap1, leading to the release and nuclear translocation of Nrf2, which enhances the expression of antioxidant response element (ARE)-driven genes. This activation helps in reducing oxidative stress and promoting cellular defense mechanisms.” (P534), but it is difficult for readers to judge it without references as evidence for each description. Authors should add references for these descriptions.
Author Response
We thank the reviewer for evaluating the manuscript and providing feedback.
Q1. First, authors suggest “Inclusion criteria for the study encompassed individuals of both genders who had a confirmed diagnosis of hip and knee osteoarthritis (ICD-10 codes M16 and M17) and type 2 diabetes mellitus (ICD-10 code E11).” (L121), but they do not explain the details of how to diagnoses hip and knee osteoarthritis and type 2 diabetes mellitus (e.g., levels of serum glucose and HbA1c). Without the detailed descriptions, it is difficult for readers to understand what they did. Authors should add detailed descriptions regarding how to diagnoses hip and knee osteoarthritis and type 2 diabetes mellitus.
A1. The authors acknowledge the importance of providing detailed diagnostic criteria for hip and knee osteoarthritis and type 2 diabetes mellitus. In our study, the diagnosis of osteoarthritis was confirmed through clinical assessments, including radiographic imaging and standardized clinical criteria. At the same time, type 2 diabetes mellitus was diagnosed based on elevated serum glucose levels and HbA1c values, as outlined by the American Diabetes Association. To enhance clarity and address these points, we have now revised the manuscript.
Q2. Second, authors showed odds of OA+T2DM for HOMA-IR, HbA1C, and serum IgA (table 7 and Figure 3), but it is difficult to understand why they showed this analysis. Authors may calculate propensity scores, but they did not use them in main analyses (i.e., table 8-11). Moreover, it is difficult to understand what they did in table 7 and figure 3 (e.g., what is “group” in title of table 7? and what is population authors used in these analyses? What did authors adjust for?). Authors should explain why they showed these analyses and what they did in this manuscript.
A2. Based on our statistical analysis of potential predictors that may influence the development of comorbid osteoarthritis and type 2 diabetes mellitus in patients, we identified the indicators mentioned in this section of the manuscript. To enhance the clarity and conciseness of the article, we have now removed the figures associated with this analysis.
Q3. Third, authors showed intervention using Alpha-Lipoic Acid, but they do not show difference of difference. Authors showed “statistical difference between groups before treatment” and “statistical difference between groups after treatment,”, but these results may be biased due to small sample size and skewed distributions. Authors should add results regarding difference of difference in table 8-11.
A3. We have now conducted an additional Difference-in-Differences (DiD) test to further clarify the impact of the intervention using Alpha-Lipoic Acid. The results of the analysis, along with the statistical significance, have now been incorporated into Tables 8-11 for better clarity and accuracy.
Q4. Fourth, authors do not randomize the intervention group, which may lead to biased results. Authors should add limitation regarding his issue.
A4. We have now incorporated this observation regarding the lack of randomization in the intervention group into the limitations section of the manuscript. A CONSORT diagram has also been included to improve the accessibility and understanding of our research.
Q5. Finally, authors described some of sentences without citation or justification as follows; “OA is a degenerative joint disorder that primarily targets the articular cartilage and is influenced by factors such as trauma, metabolic processes, and comorbidities.” (P44), “standard protocols” (P114), “Alpha lipoic acid (ALA) reduces the formation of reactive oxygen species (ROS) by directly scavenging free radicals, including hydroxyl radicals and hydrogen peroxide. Additionally, ALA chelates transition metal ions, preventing the conversion of weak oxidants into more harmful reactive species. ALA also enhances the levels of intracellular antioxidants such as glutathione, which further helps in neutralizing ROS. Moreover, ALA can regenerate other antioxidants like vitamin C and vitamin E, thereby maintaining a robust antioxidant defense system within the cell. ” (P511), “Alpha lipoic acid (ALA) has been shown to inhibit the NF-κB signaling pathway in various ways. Firstly, ALA pretreatment of human large artery endothelial cells (HAEC) significantly inhibited NF-κB binding activity induced by TNF-α in a dose-dependent manner, suggesting a metal chelation effect rather than a general antioxidation effect. Secondly, ALA has been demonstrated to block NF-κB activation, which is crucial for preventing the spread of influenza virus by inhibiting the virus-induced increase in NF-κB and caspase activities. Lastly, ALA can reduce the production of pro-inflammatory cytokines by negatively regulating NF-κB, thereby modulating the immune response and reducing inflammation.(center) Alpha lipoic acid (ALA) activates the AMPK pathway, which in turn inhibits the mTOR signaling pathway, leading to reduced cell growth and proliferation. Additionally, ALA-induced AMPK activation results in the phosphorylation and inhibition of mTOR, thereby promoting autophagy and improving cellular energy metabolism. (right) The Keap1-Nrf2 pathway is a signaling pathway involved in cellular responses to oxidative stress. The regulatory protein Keap1 regulates the activity of the transcription factor Nrf2. Under conditions of oxidative stress, Nrf2 is released from Keap1 and translocates to the nucleus, where it activates the expression of genes encoding antioxidant enzymes. Alpha lipoic acid (ALA) activates the Keap1-Nrf2 signaling pathway by modifying Keap1, leading to the release and nuclear translocation of Nrf2, which enhances the expression of antioxidant response element (ARE)-driven genes. This activation helps in reducing oxidative stress and promoting cellular defense mechanisms.” (P534), but it is difficult for readers to judge it without references as evidence for each description. Authors should add references for these descriptions.
A5. The authors acknowledged the importance of providing citations to support the statements made in the manuscript. We have now added references for each description to enhance the credibility and clarity of our findings.
We thank again the reviewer for the feedback
Round 2
Reviewer 1 Report
Comments and Suggestions for Authors
The manuscript has been considerably improved and streamlined
Author Response
We thank the reviewer for the evaluation and comment
Reviewer 2 Report
Comments and Suggestions for Authors
The manuscript was improved.
Author Response
We thank the reviewer for the feedback
Reviewer 3 Report
Comments and Suggestions for Authors
Authors revied the manuscript, but this article has not fully answered some of the questions due to insufficient description.
In fact, as mentioned in the previous review, authors showed odds of OA+T2DM for HOMA-IR, HbA1C, and serum IgA (table 7 and Figure 3), but it is difficult to understand why they showed this analysis and what they did in table 7 and figure 3 (e.g., what is “group” in title of table 7? and what is population authors used in these analyses? What did authors adjust for?). Authors suggest “To enhance the clarity and conciseness of the article, we have now removed the figures associated with this analysis.”, but they do not remove figure 2 as well as table 7 and their descriptions (i.e., section 3.3). Authors should remove these descriptions as well as table 7 and figure 2, if they decided to remove them.
Author Response
We thank the reviewer for the evaluation and additional suggestions.
Reviewer's comments.
Authors revied the manuscript, but this article has not fully answered some of the questions due to insufficient description.
In fact, as mentioned in the previous review, authors showed odds of OA+T2DM for HOMA-IR, HbA1C, and serum IgA (table 7 and Figure 3), but it is difficult to understand why they showed this analysis and what they did in table 7 and figure 3 (e.g., what is “group” in title of table 7? and what is population authors used in these analyses? What did authors adjust for?). Authors suggest “To enhance the clarity and conciseness of the article, we have now removed the figures associated with this analysis.”, but they do not remove figure 2 as well as table 7 and their descriptions (i.e., section 3.3). Authors should remove these descriptions as well as table 7 and figure 2, if they decided to remove them.
Response
In accordance with the recommendations, we have now carefully reviewed the manuscript and made additional and necessary adjustments. We have removed the figures and descriptions associated with the analysis, including Figure 2 and Table 7, and we have completely eliminated section 3.3 from the article to enhance clarity and conciseness as intended.
Round 3
Reviewer 3 Report
Comments and Suggestions for Authors
Authors revied the manuscript, but this article has not fully answered some of the questions due to insufficient description.
As mentioned in the first review, authors do not randomize the intervention group, which may lead to biased results, and authors added the relevant descriptions as follows; “The lack of randomization in the intervention group may introduce bias into the results, highlighting a limitation of this study that should be considered in future research.” (L538). However, authors suggest that they randomized 71 participants in figure 1. Authors should check the manuscript, carefully.
Author Response
Dear Reviewer,
Thank you for your valuable feedback and suggestions. We have revised our article in accordance with your recommendations.
- Authors revied the manuscript, but this article has not fully answered some of the questions due to insufficient description.
As mentioned in the first review, authors do not randomize the intervention group, which may lead to biased results, and authors added the relevant descriptions as follows; “The lack of randomization in the intervention group may introduce bias into the results, highlighting a limitation of this study that should be considered in future research.” (L538). However, authors suggest that they randomized 71 participants in figure 1. Authors should check the manuscript, carefully.
We sincerely thank you for your careful comments and valuable suggestions. We apologize for the oversight in our manuscript and are deeply grateful for your thorough review. After careful consideration, we have addressed your concerns. Specifically, we clarified the absence of randomization of participants in our study, and this information has been included in the limitations section. The CONSORT diagram has been removed, and the corresponding description has been updated accordingly. Your feedback has undoubtedly improved the quality of our work, and for this, we are sincerely thankful.
Once again, thank you for your valuable recommendations.